# Remote Blood Pressure Estimation via the Spatiotemporal Mapping of Facial Videos

**DOI:** 10.3390/s23062963

**Published:** 2023-03-09

**Authors:** Yuheng Chen, Jialiang Zhuang, Bin Li, Yun Zhang, Xiujuan Zheng

**Affiliations:** 1Department of Automation, College of Electrical Engineering, Sichuan University, Chengdu 610065, China; 2Key Laboratory of Information and Automation Technology of Sichuan Province, Chengdu 610065, China; 3School of Computer Science, Northwest University, Xi’an 710069, China; 4School of Information Science and Technology, Xi’an Jiaotong University, Xi’an 710049, China

**Keywords:** blood pressure, remote estimation, spatiotemporal map, oversampling training strategy, facial video

## Abstract

Blood pressure (BP) monitoring is vital in daily healthcare, especially for cardiovascular diseases. However, BP values are mainly acquired through a contact-sensing method, which is inconvenient and unfriendly for BP monitoring. This paper proposes an efficient end-to-end network for estimating BP values from a facial video to achieve remote BP estimation in daily life. The network first derives a spatiotemporal map of a facial video. Then, it regresses the BP ranges with a designed blood pressure classifier and simultaneously calculates the specific value with a blood pressure calculator in each BP range based on the spatiotemporal map. In addition, an innovative oversampling training strategy was developed to handle the problem of unbalanced data distribution. Finally, we trained the proposed blood pressure estimation network on a private dataset, MPM-BP, and tested it on a popular public dataset, MMSE-HR. As a result, the proposed network achieved a mean absolute error (MAE) and root mean square error (RMSE) of 12.35 mmHg and 16.55 mmHg on systolic BP estimations, and those for diastolic BP were 9.54 mmHg and 12.22 mmHg, which were better than the values obtained in recent works. It can be concluded that the proposed method has excellent potential for camera-based BP monitoring in the indoor scenarios in the real world.

## 1. Introduction

Blood pressure (BP) is a primary physiological parameter of the human body and is an essential basis for disease diagnosis. Cardiovascular diseases are the leading cause of death globally. Most of them are highly related to hypertension. In this context, convenient ordinary monitoring of blood pressure in daily life becomes critical, especially for long-term bedridden patients, sedentary office workers, etc. Many outstanding works have successfully calculated BP values by extracting the morphology and temporal characteristics of the blood volume pulse (BVP) signals obtained with the photoplethysmography (PPG) technology [1,2]. A continuous BP waveform could also be recovered according to the BVP signals [3]. These works proved that BVP signals can effectively reflect the changes in blood flow in the human cardiovascular system and further reflect changes in BP. However, photoelectric contact sensors are needed to obtain BVP signals, which could limit their practicality in situations such as intensive care for severely burned/wounded patients or premature infants or when free movement is required. Recently, many excellent works were proposed that could robustly extract BVP signals and accurately calculate physiological parameters based on facial videos [4,5]. Among these works, some studies [6,7,8] indicated that spatiotemporal maps of facial videos could exploit blood-flow states in blood vessels. These previous results suggest that spatiotemporal maps of facial videos can be used to calculate BP values by using the hidden blood-flow information in the facial videos. Recently, remote methods that utilized facial videos to estimate BP have continually undergone research and development for improved accuracy [9].

However, the accuracy of BP estimation is easily affected by many factors, including the professional abilities of those conducting it, the height, weight, and age of the subjects, and the seasons and the environments [10]. Apart from these factors, there are many more challenges in remote BP estimation when using facial videos. For example, head motion creates difficulties for region-cutting algorithms, causes residual motion images, and degrades the image quality. In addition, a weak remote BVP signal reflecting changes in blood volume in the capillaries of the upper skin layers [11] can easily be contaminated and submerged by changes in the ambient light [5,12]. Moreover, because of the difficulties in BP data collection and the privacy issues raised by facial videos, there are very few large-scale BP databases with facial videos. The lack of data greatly hinders the rapid development of non-contact BP estimation.

Therefore, this study proposes the end-to-end non-contact BP estimation network based on facial videos. The main contributions are summarized as follows:An end-to-end network that uses spatiotemporal maps of facial videos for remote BP estimation is proposed.A BP classifier that transforms a regression problem into a joint problem of classification and regression is proposed.An oversampling training scheme for effectively addressing the unbalanced distribution of BP values in the training process is exploited.

## 2. Related Work

Blood pressure measurement can be classified into invasive and non-invasive measurement. Invasive measurement is the most accurate, but it is not suitable for daily BP measurements. It is used in critically ill patients. Non-invasive measurement indirectly measures BP by using various sensing technologies and signal processing methods. Moreover, non-invasive measurement can be divided into cuff measurement and cuff-less measurement. In general, cuff measurement is more accurate than cuff-less measurement. However, repeated cuff measurements will cause discomfort and even damage people’s bodies. For long-term BP monitoring, cuff-less BP measurements received significant attention in recent years. With the development of PPG technology, researchers have successfully calculated BP based on the features of BVP signals [13,14] or the combination of BVP and electrocardiograph signals [15]. In the time domain, pulse transit time (PTT) is one of the most commonly used features for obtaining BP [16]. PTT is usually defined as the time required for a heartbeat pulse to propagate from the heart to the body peripherals. Some studies have proved that PTT is an independent predictor of cardiovascular events that are closely related to BP [17,18]. In addition, Rohan et al. found that morphological features extracted from BVP signals can also be used to accurately calculate the arterial BP value [19], and these were obtained with the non-invasive skin-contact sensors derived from the PPG technology. The features of BVP include systolic upstroke time, diastolic time, and the time delay between the systolic and diastolic peaks of the BVP waveform. Although significant progress has been made in cuff-less BP measurement, most of the work is that of collecting PPG signals through direct or indirect contact with the human body through sensors. This contact approach causes inconvenience in measurement to a certain extent.

In recent years, remote photoplethysmography (rPPG) has been proposed as an alternative to PPG technology in some application secnarios [20,21]. A camera captures the subtle changes in skin color caused by cardiac pulsation; the rPPG technology obtains the BVP signals related to cardiopulmonary function [22] and then calculates physiological parameters such as heart rate, blood oxygen saturation, etc. Therefore, it seems reasonable to assume that if PPG signals can be utilized for BP estimation, rPPG is also feasible in this task. Some studies took videos of two parts of the body and predicted blood pressure based on the estimated PTT [23,24]. However, there are some challenges to measuring BP via videos. Firstly, motion artifacts usually contaminate the BVP signals derived from videos, leading to eliminating or shifting the heart rhythm. Secondly, PTT estimation based on rPPG technology requires multiple cameras to take videos of different human body parts, so the different light intensities on the different body parts will decrease the accuracy of PTT. On the other hand, some studies used machine learning methods to estimate BP values based on the features of the BVP signals extracted from videos [25,26,27]. Schrumpf et al. [28] used contact PPG signals to train an artificial neural network and then used transfer learning to estimate BP based on the extracted BVP signals. In addition to BVP signals, demographic features (including weight, height, and other factors) were also introduced to improve the performance in BP measurement [25].

Under these conditions, this paper proposes a non-contact BP measurement method that uses the rPPG technology. Furthermore, since the spatiotemporal mapping of facial videos has abundant physiological information related to BVP signals [6], this study aims to directly extract the features of spatiotemporal maps for BP measurement.

## 3. Method

In this section, we give detailed explanations of the proposed network for remote blood pressure estimation from facial videos. Figure 1 gives an overview of the generation of the spatiotemporal feature map slices and the blood pressure estimation network.

### 3.1. Spatiotemporal Feature Map Slices

The change in the brightness of the facial skin caused by periodic heartbeats is very subtle; thus, noise caused by movement of the face and ambient light can easily contaminate the physiological information in facial videos. Considering the goal of obtaining relatively stable pulse wave features, we propose spatiotemporal feature map slices (STSs) to highlight the physiological information in facial videos. An overview of this part is shown in Figure 2.

#### 3.1.1. Definition of Regions of Interest

In order to make full use of informative parts containing effective signals and eliminate the noisy parts, such as the eyes, nose, hair, and mouth, four regions of interest (ROIs) are defined by selecting the forehead and cheek, which have rich vascular information. First, the authors used SeetaFace6 to detect the face, localize 68 landmarks, and select seven points to delineate the ROIs in each frame in the video sequence, as shown in Figure 2. Next, we calculated the average pixel values of each ROI to represent the physiological information with Equation (Equation 1), which was noted as the initial spatiotemporal map.
(1)$$f\left(n,t\right)=\frac{\sum_{x,y\in R\left(n\right)}V\left(x,y,t\right)}{A\left(n\right)}$$
where R(n) represents the $n_{th}$ ROI, A(n) represents the number of pixels in the selected ROI, V(x,y,t) represents the pixel value of the $n_{th}$ ROI position (x,y) of frame t of the facial video sequence, and $f\left(n,t\right)$ represents the initial spatiotemporal map.

#### 3.1.2. Data Augmentation

Due to the movement of artifacts and variations in the environmental illumination during signal acquisition, the accuracy of non-contact blood pressure detection is easily affected. Therefore, a data augmentation strategy is required to solve the problem of signal contamination. In 2022, Zhuang et al. [29] proposed a modified YUV color space that gives more attention to the features of the brightness dimension in color space and reduces the noise caused by variations in the environmental illumination. First, we randomly mask a small part of the initial spatiotemporal map f(n,t) along both the time dimension and spatial dimension, as shown in Figure 2. Then, the modified YUV color space is used to extract information related to BVP to get $g\left(n,t\right)$. In the BP estimation task, the modified YUV color space performs better than the traditional RGB and YUV color spaces. The modified YUV color space transformation can be formulated as in Equation (Equation 2).
(2)$$\left[\begin{array}{c} Y_{t}\left(x,y\right)\\ U_{t}\left(x,y\right)\\ V_{t}\left(x,y\right) \end{array}\right]=\left[\begin{array}{ccc} 0.299 & 0.587 & 0.114\\ -0.169 & -0.331 & 0.5\\ 0.5 & -0.419 & -0.081 \end{array}\right]\left[\begin{array}{c} R_{t}\left(x,y\right)\\ G_{t}\left(x,y\right)\\ B_{t}\left(x,y\right) \end{array}\right]$$
where *t* indicates the video frame index, $R_{t}\left(x,y\right)$, $G_{t}\left(x,y\right)$, $B_{t}\left(x,y\right)\in g_{roi}^{\prime }\left(t,r_{i}\right)$, and $R_{t}\left(x,y\right)$, $G_{t}\left(x,y\right)$, $B_{t}\left(x,y\right)$ represents the pixel value in the RGB color space, while $Y_{t}\left(x,y\right)$, $U_{t}\left(x,y\right)$, $V_{t}\left(x,y\right)$ are the pixel values of the modified YUV color space. All of them constitute the video sequence after data enhancement. The enhanced video sequence is denoted as $g\left(n,t\right)$.

#### 3.1.3. Spatiotemporal Slicer

Noise caused by facial movement and ambient light causes pulse wave distortion over a certain period of time. Time slicing allows a network to eliminate feature maps with great amounts of noise; thus, we propose spatiotemporal slicer to get stable pulse wave feature maps. For the $t^{th}$ frame of the enhanced video sequence $g\left(n,t\right)$, we firstly get a set of *n* ROIs of face $R_{n}$ = $\left\{R_{1},R_{2},\ldots ,R_{n}\right\}$ and put them side by side, as shown in Figure 2. Then, we calculate the average pixel values of each color channel (the modified YUV color space) for all of the non-empty subsets of $R_{n}$. The whole procedure can be written as:(3)$$SS\left(t\right)=\mathrm{CAT}\left(\emptyset \left(\overset{N}{\underset{n=1}{⋃}}g\left(n,t\right)\right)\right)$$
where $\overset{N}{\underset{n=1}{⋃}}g\left(n,t\right)$ represents putting the enhanced video sequence $g\left(n,t\right)$ of each ROI side by side, ∅(·) represents the operation of extracting non-empty subsets, and CAT(·) represents the feature vector splicing operation. After the above operations, we initially extracted the physiological features in each frame into the spatiotemporal feature maps. Then, we divided them into multiple parts with sliding windows with a step length of *L* because the length of the whole video was too large to be put into the network. All separate time–space fragments $\left\{T_{1},T_{2},\ldots ,T_{M}\right\}$ were constructed as spatiotemporal feature map slices $STS\left(m\right)$, which could be defined as:(4)$$STS\left(m\right)=\overset{(N+1)\times cl}{\underset{t=N\times cl}{⋃}}SS\left(t\right)$$
where *m* is the index of STS, and cl is the length of the sliding windows. In order to use video information effectively and help the network learn, we normalized the pixel values of all video frames to the range of [0, 1].

### 3.2. Network Architecture

After we generated the STSs, we could further use these informative representations for blood pressure estimations. However, some of these hand-crafted STSs were polluted by great amounts of noise, such as head movements and changes in illumination conditions. In order to solve this problem, we propose the blood pressure estimation network (BPE-Net). In the proposed BPE-Net, a residual convolution neural network [30] is used as the backbone network.

Specifically, as shown in Figure 1, a feature extraction network module is used to perform high-dimensional feature extraction on the $STS\left(m\right)$. Then, the results are fed to the subsequent LSTM to strengthen the temporal correlation and obtain the high-dimensional semantic features *F*. Then, a blood pressure classifier is used to classify the high-dimensional semantic features *F* to obtain the blood pressure interval. This operation can establish a benchmark for the features and effectively reduce the overfitting phenomenon in the network. Finally, the results of the classifier are integrated with the output of the feature extractor to calculate the specific value of the blood pressure, which can also effectively help in classifier training.

We express the whole blood pressure estimation process with the following formula:(5)$$F=\mathrm{CAT}\left(⋃\mathrm{FE}\left(STS(m)\right)\right)$$
(6)$$F_{cla}=\mathrm{CLA}\left(F\right)$$
(7)$$R_{reg}=\mathrm{REG}\left(\mathrm{CAT}\left(F,F_{cla}\right)\right)$$
(8)$$R_{cla}=\mathrm{SOFTMAX}\left(F_{cla}\right)$$
where FE represents the feature extraction network composed of the depth residual network and LSTM, CAT(·) represents the dimension splicing operation, CLA(·) represents the blood pressure classifier, REG(·) represents the blood pressure calculator, $R_{cla}$ and $R_{reg}$ represent the output results of the classifier and calculator, respectively, and SOFTMAX(·) represents a softmax operation. Here, the blood pressure classifier and blood pressure calculator are two fully connected layers. The input and output of the CLA that is used for classification are 64-dimension feature vectors and four-dimension vectors, respectively. Those of the REG are 68-dimension vectors and the blood pressure values, respectively.

In addition to designing the network architecture, we also need an appropriate loss function to guide the networks. We used the cross entropy loss function to train the classifier and the L1 loss function to train the calculator. The two loss functions were combined to train the proposed model, and they were formulated as follows:(9)$$Loss_{cla}=\frac{1}{N}\sum_{i}-\left[q_{i}\cdot \log\left(p_{i}\right)+\left(1-q_{i}\right)\cdot \log\left(1-p_{i}\right)\right]$$
(10)$$Loss_{reg}=\frac{\sum_{i=1}^{N}\left|f_{i}-h_{i}\right|}{N}$$
(11)$$Loss=Loss_{cla}+Loss_{reg}$$
where $p_{i}$ represents the result of the classifier, $q_{i}$ represents the category label of the true value of the blood pressure, $f_{i}$ represents the result of the calculator, and $h_{i}$ represents the true value of the blood pressure. The final output result R needs to be calculated with the classifier result as the reference value and the deviation calibration of the calculator.
(12)$$R=\alpha \cdot \mathrm{STA}\left(R_{cla}\right)+\beta \cdot R_{reg}$$
where STA represents the reference value of each blood pressure interval and the deviation weight, α represents the weight of the calculator’s results, and β represents the weight of the classifier’s results.

### 3.3. Oversampling Training Strategy

In order to deal with the unbalanced distribution of data in different blood pressure intervals in databases, an oversampling training strategy is proposed. First, according to the distribution and the level of blood pressure, we used different grouping strategies to divide systolic and diastolic pressure data. So, we divided the blood pressure data into four groups based on the values of the blood pressure. Then, we extracted the data from each blood pressure group in order to ensure that equal numbers of samples in each batch were from each blood pressure range. In other words, the ratio of samples from each group in each batch was 1:1:1:1. It is worth noting that we extracted only one sample from one subject. For each batch, four-fifths of the samples were used as the training set, and one-fifth of the samples were used as the validation set.

## 4. Experiments and Results

In this section, we provide evaluations of the proposed method, including cross-database testing and an ablation study.

### 4.1. Datasets and Experimental Settings

#### 4.1.1. Datasets

We evaluated our method on a public dataset and a private dataset.

**MMSE-HR** [31] is a public non-contact heart rate and blood pressure estimation dataset composed of 102 facial videos from 40 subjects (no makeup), which were recorded at 25 frames per second (fps). A physiological data acquisition system was used to collect the average HR and BP values. The MMSE-HR dataset involves a wide range of ages, diverse ethnic/racial ancestries, and various facial expressions and head movements. Therefore, the MMSE-HR dataset is popular in the area of blood pressure estimation, and we chose it as our validation dataset.

**Multimodal Physiological Measurement—Blood Pressure (MPM-BP)** is a private dataset for non-contact blood pressure estimation. In this private MPM-BP dataset, the physiological data and corresponding facial videos of 132 people aged 18–24 were collected by using a multi-channel physiological signal acquisition system (Biopac M160), a blood pressure monitor (OMRON HEM-1020), and a mobile phone camera (Huawei Mate30) without a beauty function. Biopac M160 ran the entire time during the data collection. When the mobile phone started to capture video information, Biopac M160 had a corresponding timestamp, which was used to synchronize the physiological signals and video signals. The facial video for each subject was recorded for about 1 min. The frame rate of the videos was 30 fps, the resolution was 1920 × 1080, and the distance between the camera and subjects was about 0.5 m. In addition, the blood pressure signals were collected at the same time as the video signals. The distributions of systolic blood pressure (SBP) and diastolic blood pressure (DBP) for the MPM-BP dataset are shown in Figure 3. Moreover, the devices and setup are illuminated in Figure 4.

#### 4.1.2. Evaluation Metrics

Three metrics were used to evaluate the performance of the network, namely, the standard deviation (SD), the root mean square error (RMSE), and the mean absolute error (MAE).

#### 4.1.3. Training Details

For all of the experiments, the facial videos were sampled to 30 fps, the size of all convolution kernels was 3 × 3, the step was 1, and the feature extraction network corresponding to each time domain segment used the same parameters. All of the networks were implemented with PyTorch and trained with NVIDIA-v100. The ADAM optimizer with an initial learning rate of 0.001 was used for training. The maximum epoch number for training was set to 30 for the experiments on the MPM-BP dataset and to 30 for the experiments on the MMSE-HR dataset.

### 4.2. Ablation Study

We performed an ablation study of our proposed method. All ablation experiments were performed on the MPM-BP dataset using the five-fold cross-validation method.

#### 4.2.1. Impact of the Modified YUV Color Space

As mentioned before, our experiments proved that the modified YUV color space performed better in blood pressure estimation. As shown in Table 1, in terms of systolic blood pressure estimation, there was an increase between RGB and modified YUV from 8.42 mmHg to 8.07 mmHg in the MAE, from 10.34 mmHg to 9.94 mmHg in the RMSE, and from 10.34 mmHg to 9.81 mmHg. Similarly, in the diastolic pressure estimation task, the modified YUV color space still performed better than the traditional RGB and YUV color spaces. The modified YUV color space increased the weight of the Y channel, i.e., brightness, which was the parameter that best visualized the intensity of the heart rhythm hidden in the skin regions in the videos.

#### 4.2.2. Impact of the Spatiotemporal Slicer

In order to verify the effectiveness of the combination of the spatiotemporal feature map slices and LSTM, we trained our network by using different slice lengths (90, 150, 225, and 450 frames). In the training, we used the oversampling training scheme on systolic and diastolic blood pressure data. Moreover, we used ResNet18 as the backbone network and modified YUV as the color space. In this experiment, we proved that the specific length of the spatiotemporal slicer would greatly affect the final training results, and the optimal scheme of spatiotemporal slicing can be found in the following table.

As shown in Table 2, the spatiotemporal slicer with LSTM could help the network obtain the detailed characteristics of physiological information more effectively, and the slice length of 150 frames could maximize the effect of this network combination. Specifically, the MAEs for systolic blood pressure and diastolic blood pressure reached 8.07 mmHg and 6.78 mmHg, respectively. Compared with other slice length schemes, the MAEs of systolic blood pressure and diastolic blood pressure increased by 0.37 mmHg and 0.55 mmHg, respectively, which was much better than the network scheme without temporal and spatial slices (15.78 mmHg and 15.04 mmHg, respectively). For SD and RMSE, in practical measurement, the number of observations n is always limited, and the actual value can only be replaced by the most reliable (optimal) value. The RMSE is susceptible to a group of significant or minor errors in measurement, so it can reflect measurement precision well. It can be seen from the table that the RMSE and SD of the 150-slice length scheme were still the best. The reason may be that a 150-frame slice contained enough physiological information to calculate the BP values for the network. However, with the increase in the slice length, the slice contained more non-physiological information. The ability of the network to extract physiological features would also be affected. This also explains why the precision decreased with the increase in the slice length.

#### 4.2.3. Impact of the Oversampling Strategy

With the standard data sampling training scheme as a comparison (in each epoch, training samples were imported into the network for training), we tested the effectiveness of the oversampling training scheme. The above results show that the combination of spatiotemporal slices and LSTM was the best when the slice length was 150. Therefore, in this test, a slice length of 150 was used as the primary network structure. Table 3 shows that the training scheme enabled the same network model to have more vital fitting abilities and robustness. The MAEs for systolic blood pressure and diastolic blood pressure decreased from 8.27 mmHg and 6.92 mmHg to 8.07 mmHg and 6.78 mmHg. The RMSE also decreases by 0.2 mmHg and 0.15 mmHg, respectively. This shows that for the blood pressure task with few training samples, an appropriate oversampling training scheme is helpful for further enhancing the network’s potential.

#### 4.2.4. Impact of the Loss Function

Given that systolic blood pressure ranges from 90 mmHg to 160 mmHg with a maximum span of 70 mmHg, while diastolic blood pressure ranges from 50 mmHg to 100 mmHg with a span of only 50 mmHg, the difference in the difficulty levels between the two target tasks had to be considered when selecting the most suitable loss function. Two standard regression loss functions, L1 and L2, were tested.
(13)$$L1=\sum_{i=1}^{n}\left|y_{i}-f\left(x_{i}\right)\right|$$
(14)$$L2=\sum_{i=1}^{n}{\left(y_{i}-f\left(x_{i}\right)\right)}^{2}$$

As shown in Table 4, L1 was better than the L2 loss function in predicting BP, as the MAE, RMSE, and SD were all the lowest. This may be because BP has an extensive range and many extreme values. When L2 was used as the loss function, the model minimized the error caused by extreme values, giving them more weight, which impacted the overall effect of the model. Compared with L2, L1 had better performance when the data were not favorable to the extreme value of the predicted results. In general, L1 was able to overcome the adverse effects of extreme values.

### 4.3. Cross-Dataset Testing

In the cross-dataset testing, all 132 MPM-BP data samples were used to training the proposed BPE-Net, and then the MMSE-HR dataset was used for testing. The sample ratio of the training dataset and testing dataset was close to 1:1. Recently, some studies provided solutions for blood pressure estimation based on rPPG [26,28]. In this study, we reproduced the non-contact blood pressure implementation system (NCBP) [26] algorithm, trained it with MPM-BP, and tested it on MMSE-HR. Moreover, the results of the non-invasive blood pressure prediction (NIBPP) method [28] were also used for comparison. The paper on NIBPP [28] only showed the MAE results. The comparative results are listed in Table 5. After fine-tuning, the proposed BPE-Net obtained the best performance. The SD, MAE, and RMSE for systolic blood pressure were 16.02 mmHg, 16.55 mmHg, and 13.6 mmHg, respectively, and those for diastolic blood pressure were 11.98 mmHg, 12.22 mmHg, and 9.54 mmHg, respectively. The results show that the 150-slice length scheme achieved the best results in cross-dataset testing. Therefore, the proposed BPE-Net achieved better performance than that of other the recent studies. We can conclude that the proposed BPE-Net with a 150-slice length scheme can provide relatively accurate blood pressure estimation from facial videos.

## 5. Conclusions

This paper presents a novel non-contact end-to-end blood pressure estimation network that only uses facial video information to quickly calculate diastolic and systolic blood pressure in 15 s. In order to compare our results to those of previous work, we performed experiments on the public MMSE-HR dataset and the self-collected MPM-BP dataset, and we additionally evaluated the mean performance. The proposed BPE-Net achieved a proportion of the difference between the predicted and actual blood pressure values of less than 10 mmHg is 74.2% and 78.8% for SBP and DBP, respectively. The results did not meet the requirements defined in the relevant BHS and AAMI standards [32], which require the probability of a BP measurement device to provide an acceptable error (BP < 10 mmHg) to exceed 85%. It is a challenge to achieve the precision of contact blood pressure measurement equipment in the field of remote blood pressure estimation. Our method provides a solution for remote blood pressure estimation and can estimate blood pressure relatively accurately with mobile phones. In addition, it is worth noting that all experiments with this method were carried out indoors. The applicability of this method in different environments (such as with different lights, different ages, or while wearing makeup or not) needs to be further verified in the future.

This paper innovatively proposed a BP classifier to transform a regression problem into a joint problem of classification and regression. We found that the predicted SBP range of 110–130 mmHg and the DBP range of 60–80 mmHg were more accurate than other BP ranges. The reason may be that the distribution of BP in the MPM-BP dataset was unbalanced, and we repeatedly took samples from the BP groups with ranges of 110–130 mmHg and 60–80 mmHg. Moreover, other factors, such as age and skin status, will also affect the results. Therefore, it is necessary to expand the diversity of the dataset and improve the classification ability of the network for different BP intervals in the future.

NCBP is a manual feature extraction method, and our results show that the NCBP’s error was significantly higher than those of the two end-to-end methods. NIBPP’s MAE was similar to that of our method with 90 or 225 frame slices. However, the MAE of our method with 150 frame slices was much lower than that of NIBPP. This shows that the length of the spatiotemporal slices that we adopted allowed the preliminary extraction of physiological signals to be realized without too many non-physiological signals.

## Figures and Tables

**Figure 1 sensors-23-02963-f001:**
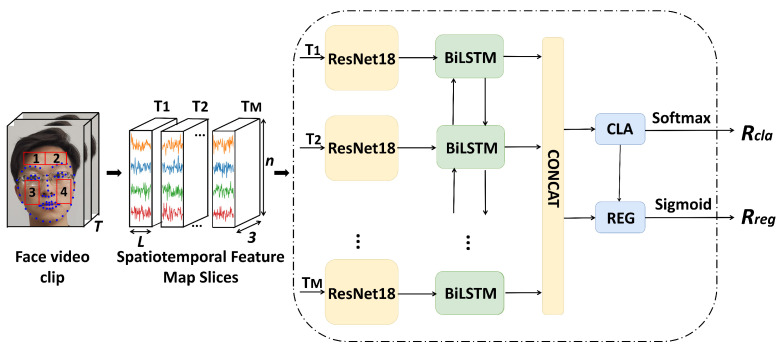
An overview of our blood pressure estimation network. We first generate the corresponding spatiotemporal feature map slices of the input facial video clips. Then, a feature extractor composed of a depth residual network and BiLSTM fits a high-dimensional feature, which is fed into the blood pressure classifier to locate the blood pressure interval through this feature. Finally, the blood pressure calculator combines the results of the feature extractor and the blood pressure classifier to output the final blood pressure value.

**Figure 2 sensors-23-02963-f002:**
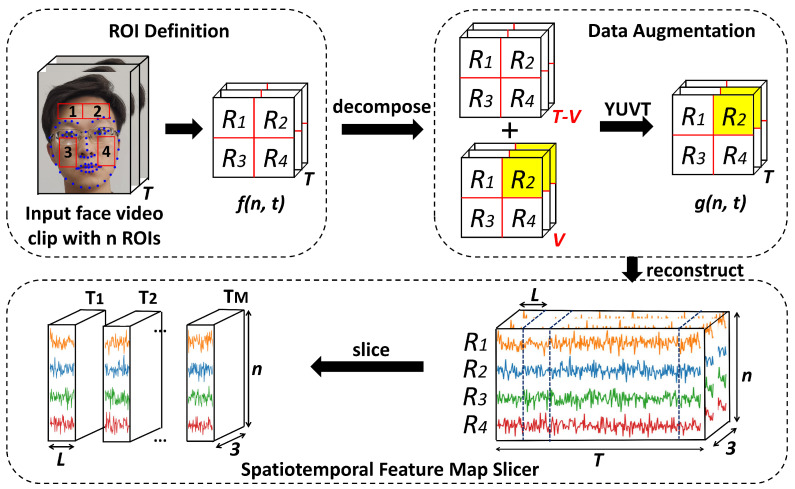
An illustration of the generation of spatiotemporal feature map slices from an input facial video clip of T frames. Firstly, we use Seetaface6 to detect the human face, localize 68 landmarks, and define four ROIs; then, we calculate the average pixel values of each ROI to get spatiotemporal maps. Secondly, the data augmentation module randomly masks a part of the cropped spatiotemporal maps presented above along both the time domain and the spatial domain; then, they are transformed from the RGB color space into the modified YUV color space. Finally, the cropped video sequence is sliced to generate the spatiotemporal feature map slices for the following network.

**Figure 3 sensors-23-02963-f003:**
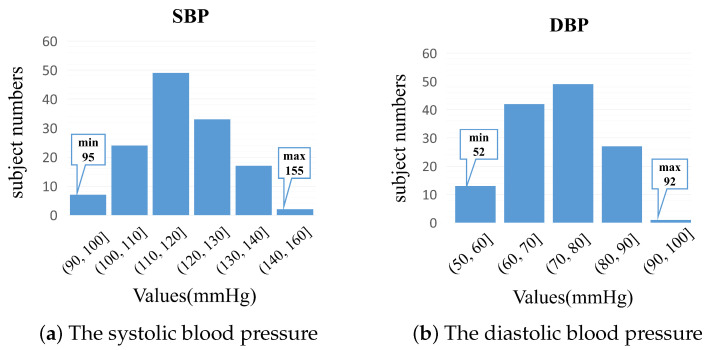
The distributions of the ground-truth blood pressure values in the MPM-BP dataset.

**Figure 4 sensors-23-02963-f004:**
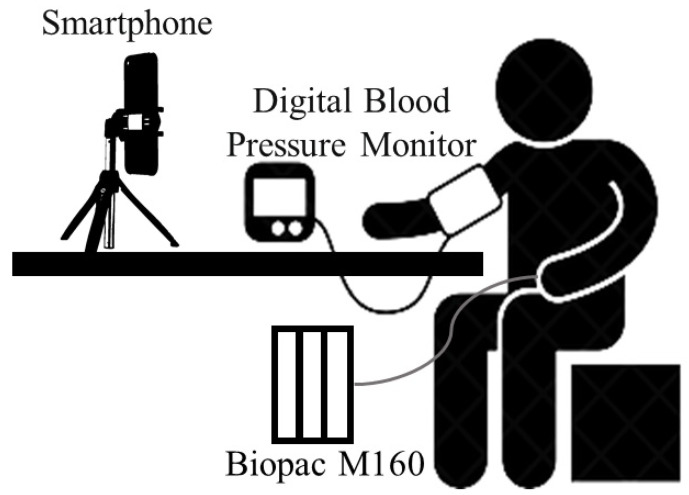
Devices and setup used to collect MPM-BP.

**Table 1 sensors-23-02963-t001:** Comparison of the performance of different color spaces.

	Color Space	SD (mmHg)	RMSE (mmHg)	MAE (mmHg)
SBP	RGB	10.13	10.34	8.42
	YUV	9.99	10.18	8.21
	modified YUV	9.81	9.94	8.07
DBP	RGB	8.58	8.82	7.12
	YUV	8.41	8.62	6.97
	modified YUV	8.28	8.45	6.78

Notes: Red indicates the best performance of different color spaces. Blue indicates the second-best performance of different color spaces.

**Table 2 sensors-23-02963-t002:** Comparison of the performance of different clip lengths.

	Method	SD (mmHg)	RMSE (mmHg)	MAE (mmHg)
SBP	w/o slicing	10.49	17.85	15.78
	slice-90	10.06	10.33	8.44
	slice-150	9.81	9.94	8.07
	slice-225	10.09	10.16	8.32
DBP	w/o slicing	9.28	17.06	15.04
	slice-90	8.35	9.12	7.31
	slice-150	8.28	8.45	6.78
	slice-225	8.33	9.05	7.33

Notes: Red indicates the best performance of different slice length schemes. Blue indicates the second-best performance of different slice length schemes.

**Table 3 sensors-23-02963-t003:** Comparison of the performance of different networks.

	Method	SD (mmHg)	RMSE (mmHg)	MAE (mmHg)
SBP	w/o OSS	10.12	10.14	8.27
	w/ OSS	9.81	9.94	8.07
DBP	w/o OSS	8.35	8.6	6.92
	w/ OSS	8.28	8.45	6.78

Notes: Red indicates the best performance of different networks.

**Table 4 sensors-23-02963-t004:** Comparison of the performance of different loss functions.

	Loss Function	SD (mmHg)	RMSE (mmHg)	MAE (mmHg)
SBP	L1	9.81	9.94	8.07
	L2	10.65	10.63	8.63
DBP	L1	8.28	8.45	6.78
	L2	9.03	9.24	7.34

Notes: Red indicates the best performance of different loss functions.

**Table 5 sensors-23-02963-t005:** Results of BP estimations in the MMSE-HR dataset.

	Method	SD (mmHg)	RMSE (mmHg)	MAE (mmHg)
SBP	NCBP [26]	19.81	22.43	17.52
	NIBPP [28]	−	−	13.6
	BPE-Net (90)	17.02	17.35	13.42
	BPE-Net (150)	16.02	16.55	12.35
	BPE-Net (225)	16.98	17.12	13.15
DBP	NCBP [26]	15.21	15.23	12.13
	NIBPP [28]	−	−	10.3
	BPE-Net (90)	13.34	13.44	10.41
	BPE-Net (150)	11.98	12.22	9.54
	BPE-Net (225)	12.87	13.22	10.33

Notes: Red indicates the best performance of different methods. Blue indicates the second-best performance of different methods. BPE-Net (90) indicates BPE-Net with the 90-slice length scheme.

## Data Availability

The data are not publicity avaliable due to privacy or ethical restrictions.

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
