# Peer review of "Remote Blood Pressure Estimation via the Spatiotemporal Mapping of Facial Videos"

_sensors, 2023, doi:10.3390/s23062963_

Round 1

Reviewer 1 Report

(1)The abstract mentions that the existing method "is inconvenient and unfriendly to the long-time BP measurement.", so can the method described in this paper effectively solve this problem?

(2)The conclusion of this paper mentions that It concludes that the proposed method has excellent potential for camera-based BP monitoring in realworld scenarios, however, the influence of different interference factors, such as distance, light and movement etc., are not reflected in the data set.

(3)The impact of the age range and facial skin status (sucha as wrinkles, makeup, etc.) of the people in the public data set is not mentioned. 

(4)According to the detection method shown in Figure 4, the problem of long-term blood pressure measurement cannot be solved. In addition, what are the specific requirements for the environment when sampling? Can stable blood pressure measurements be obtained in any environment?

(5)Do you use flash, beauty and other functions when taking pictures with the Huawei camera? Will the quality of different mobile phone cameras affect the results?

(6)The paper applied an oversampling strategy to solve the related problems, but did not explain why slice-150 performs better than slice-225?

Author Response

Dear reviewer,

Thank you very much for your comments and professional advice. We have studied all comments carefully and have made conscientious correction. Because the website can only submit one pdf file, the revised paper cannot be uploaded together with our responds to the comments. We've sent the revised paper to the editor Ms. Eliza Wu. Please see the attachment.

Thank you very much for your attention and time.

Best wishes,

Yuheng Chen

Author Response

(The authors gave the same response as above.)

Round 2

Reviewer 1 Report

The revision for the paper is appropriate.

Author Response

Dear reviewer,

Thank you very much for your positive and constructive comments again.

Best Wishes,

Yuheng Chen

Author Response

Dear reviewer,

Thank you very much for your positive and constructive comments and suggestions. We have revised the paper according to the comments. Please see the attachment.

Thank you again for taking your time to review this manuscript. 

Best Wishes,

Yuheng Chen
